# Determinants of Residents' E-Waste Recycling Behavioral Intention: A Case Study from Vietnam

**Hong Thi Thu Nguyen [1,2], Rern-Jay Hung [3,*], Chun-Hung Lee [4]  and Hang Thi Thu Nguyen [5]**

1.  Department of Tropical Agriculture and International Cooperation, National Pingtung University of Science and Technology, Pingtung 91201, Taiwan; hongbkes@gmail.com
2.  Faculty of Chemistry, The University of Danang - University of Science and Education, Le Duan Road, Danang 550000, Vietnam
3.  International Bachelor Degree Program in Finance, National Pingtung University of Science and Technology, Pingtung 91201, Taiwan
4.  Department of Natural Resources and Environmental Studies, National Dong Hwa University, Hualien 97401, Taiwan; chlee@gms.ndhu.edu.tw
5.  Faculty of Fundamental Science, Danang University of Medical Technology and Pharmacy, Hung Vuong Road, Danang 550000, Vietnam; thuhang119@gmail.com
*   Correspondence: bruce@mail.npust.edu.tw; Tel.: +886-8-770-3202

**Abstract:** An enormous volume of electronic waste (e-waste) is currently being generated in Vietnam, threatening to render this country as an e-waste dumping region. Although the residents play an indispensable role in the e-waste management system, there is presently no or very limited studies available which involve public perceptions on the e-waste recycling in Vietnam. In this study, based on the theory of planned behavior (TPB), the structural equation modeling (SEM) was employed to examine the key factors influencing e-waste recycling behavioral intention of residents in Danang city, Vietnam. Data analyzed from 520 questionnaires revealed that environmental awareness and attitude toward recycling, social pressure, laws and regulations, cost of recycling, and inconvenience of recycling significantly directly affected residents' behavioral intention, with laws and regulations being the strongest construct significantly to predict individuals' intention. Of the five above-listed constructs, only inconvenience of recycling had a negative impact on residents' recycling behavioral intention. Moreover, past experience showed the statistically significant negative effect on the inconvenience of recycling while it had no significant impact on behavioral intention. The influences of demographic variables on recycling behavioral intention were also discussed in this paper. The findings from this research may help policy-makers have a better understanding of residents' e-waste recycling intention. That is very useful in paving the way for a successful e-waste recycling and management system not only in Vietnam, but also in other countries which are suffering from the same problems of e-waste.

**Keywords:** e-waste; theory of planned behavior; recycling behavioral intention; structural equation modeling; Vietnam

## 1. Introduction

With the benefits of advancements in the area of information and communication technology (ICT), more and more people have recently entered the global information society and digital market [1]. According to Baldé [2], a huge number of people today own more than one ICT device, and the lifetime of those products such as mobile phones, computers, and other devices and equipment is becoming shorter and shorter. Consequently, this phenomenon leads to a rapidly increasing rate of generating

electrical and electronic waste (e-waste) [3,4]. E-waste, also known as waste electrical and electronic equipment (WEEE), has considerable differences in terms of definition and categorization in each area [5]. In this paper, the definition from the Step Initiative was borrowed, "E-waste is a term used to cover items of all types of electrical and electronic equipment (EEE) and its parts that have been discarded by the owner as waste without the intention of re-use." [6]. On the global scale, a statistical data showed that 44.7 million tons (Mt) e-waste were annually generated from all countries all over the world in 2016, compared to that of 43.8 Mt global quantity in 2015. In addition, the volume of e-waste is projected to peak at 52.2 Mt by 2021 [2,7].

Like other emerging countries, Vietnam is also undergoing a dramatic increase in the consumption of electrical and electronic equipment (EEE) because of the sharp rise in product demand. Along with the fastest-growing e-waste stream, its improper disposal and treatment process, especially through open burning, disposal with municipal solid waste, or informal recycling activities, are creating potential environmental threats and leading considerable risks to human health [8,9]. Duan et al. [10] reported that emerging global e-waste environmental problems resulted from crude dismantling and informal recycling in several less developed countries in Asian and African regions. Facing this urgent situation, it is necessary for the Vietnamese government to build up appropriate environmentally sound e-waste management which not only eliminates the inappropriate e-waste disposal but also controls several potentially hazardous materials contained in EEE [11].

However, e-waste management in Vietnam is currently facing actual difficulties [12,13]. Consumers' unwillingness to pay disposal fees, users' limited awareness, and the insufficiency of funds and investments are among reasons for arising serious problems of e-waste in Vietnam [14,15]. In fact, in the research indicating the shortcomings of the extended producer responsibility (EPR) implementation in Vietnam, Nguyen et al. [3] also pointed out that the unsuccessful EPR system is due to the weak awareness of public and other stakeholders. In agreement with Nguyen et al. [3], Honda et al. [16] showed that one of the important reasons for ineffective e-waste management is the lack of awareness of end users who have a tendency to reject engaging in recycling systems in case they need to pay for the recycling fee. Therefore, to fix the e-waste problems, it is urgent to focus more on the performance of the government and the behavior of residents [17]. Indeed, both the government and city councils have to perform an active role in a product take-back system and in developing recycling infrastructure in most emerging countries, including Vietnam [18]. Moreover, the achievement of a recycling program obviously relies on the attendance of residents in the society. Thus, it is undeniable that a deeper understanding of the determinants of recycling behavioral intentions would be extremely valuable.

In the last 10 years, some studies relating to e-waste issues in Vietnam have been focused on exploring different sources of e-waste generation [4]; estimating the e-waste flow in the Asian countries through illegal (Vietnam and Cambodia, Vietnam and China) [19]; investigating the structures for e-waste recycling in Vietnam [13]; and generalizing a picture of e-waste recycling in Vietnam, including the flow of e-waste, technologies used for e-waste recycling and current situation of the informal e-waste recycling [20]. Nevertheless, studies on attitudes, behaviors, and intentions toward e-waste recycling have been thrust aside. In fact, to date, there is no or very limited studies available which involve the public perception of e-waste recycling and management in Vietnam. As discussed above, without residents' behavioral intention and their willingness to take part in recycling activities, the policies regulated by the government and strategies implemented by producers cannot be practiced smoothly and effectively. Comparing to other countries such as China, Macau, Bangladesh, India, America (the case study in California), and Nigeria, there are a large number of studies on the perspective of consumers or residents involving e-waste recycling program [8,9,17,21–27]. Even in the region of South East Asia, Malaysia and Thailand, several studies analyzing the behavior and the attitudes of people toward e-waste recycling activities were also conducted [15,28].

To clarify antecedents and drivers that motivate individuals to recycle solid waste, recent studies have employed theory of planned behavior (TPB) and focused on the impact of motives namely

attitude, subjective norm, and perceived behavioral control on recycling behavior [23,29]. Moreover, several other studies have emphasized the important impacts of situational factors such as past experiences and cost of recycling on household waste recycling behavioral intention [30,31]. However, it seems to be that few studies conducted by exploring the influences of these combined factors on e-waste recycling behavioral intention. Consequently, the need for a more comprehensive picture drawing on related influencers is crucial, which can provide both decision-makers and agencies with an in-depth understanding that can be used to enhance recycling rates.

From research gaps above, it is very urgent and necessary to have a comprehensive study to help understand the social and psychological impacts on residents' behavior toward e-waste recycling, and in moving people on to other pro-environmental behaviors. The authors believe that this study is the first one which researches on residents' attitudes, behaviors, and behavioral intention for recycling e-waste in Vietnam. This study aims to analyze the fundamental psychological and demographic antecedents promoting residents' e-waste recycling behavioral intention toward formal e-waste recycling program. To be specific, the objectives of this research are:

(i) Get a better understanding about the determinants of residents' recycling behavioral intention by exploring the relationship between environmental awareness and attitude toward recycling, social pressure, laws and regulations, cost of recycling, inconvenience of recycling, and demographic variables, as well and residents' behavioral intention;

(ii) Make suggestions and recommendations for policy-drivers and organizations which the aim to encourage residents to participate in e-waste recycling activities.

To fulfill the above objectives, this research employed extension TPB focused primarily on determinants of residents' e-waste recycling behavioral intentions toward formal e-waste recycling by using structural equation modeling (SEM) as an analysis methodology.

The nearer an understanding of residents' recycling behavior is reached, the higher the possibility of achieving success is in developing educational programs, public services, and policies aiming at increasing the recycling rate. Hence, these research objectives serve not only theoretical contribution but also very practical and applied benefits. This first study may provide references and sources of information for the future researches, in terms of a theoretical concept of influencers of residents' behavior and attitudes toward e-waste recycling. Especially, by eliciting the residents' perspective toward e-waste recycling in form of pro-environmental behavior, a practical concern of enhancing further understanding about the determinants of recycling is explored, with the hope of getting a better understanding about resident actual and intended behavior. Moreover, it will help researchers and policy-makers understand about behavior and recycling intention of Vietnamese residents for e-waste recycling activities. In this way, it is a worthwhile source of reference for planning and improving e-waste recycling and management system, which can help respond urgently to the rapidly increasing amount of e-waste in the near future, while the existing system is not truly effective.

The rest of this paper is structured in the following sections. Section 2 reviews the literature on the current situation of e-waste management in Vietnam, the TPB, and the summary of factors affecting individuals' recycling behavior. Section 3 presents the theoretical framework and hypotheses development. In Section 4, the research methodology is described and study findings are discussed in Section 5. Finally, the conclusions and policy implication are presented in Section 6.

## 2. Literature Review

### 2.1. Current Situation, Policies, and Practice of E-Waste Management in Vietnam

The generation of e-waste in Vietnam is increasing rapidly, which is projected to reach 3.7 kg per capita in 2020, compared to 1.9 kg per capita in 2014 [13]. In a study focused on five large home EEE (namely, televisions, refrigerators, washing machines, personal computers, and air conditioners) in terms of e-waste flow, Nguyen et al. [4] revealed that the estimated number of discarded appliances

is about 17.2 million appliances or equivalent to 567,000 tons in 2025. The generation of e-waste is predicted to be much higher if other e-waste sources (from illegal import-export activities) can be counted. Therefore, much attention to potential e-waste generation in Vietnam is very important [32].

Faced with the problem of rapid e-waste generation, the country has established legislative and institutional groundwork related to waste management [13]. Since 2005, there has been a requirement for taking-back of discarded electronic and electric products and batteries in the Environmental Protection Law 2005—Article 67 (then substituted by Environmental Protection Law 2015) [11,20]. It can be regarded as the first milestone for the application of the EPR system on waste management in Vietnam. Till the year 2013, Vietnam had not yet completed laws and regulations officially handling with e-waste, despite the fact that there were a number of relevant Decrees [33]. Prime Minister Decision No. 50/2013/QD-TTg which dictates retrieval and disposal of discarded products was issued in 2013. Subsequently, Prime Minister's Decision No. 16/2015/QD-TTg emphasizes the regulation of retrieval and disposal of discarded products was released in the year 2015, replacing for Decision No. 50/2013/QD-TTg. It is considered as the basic legislation used for the EPR system applied on the discarded products, consisting of e-waste. It also brings vital insights for improvement of Vietnamese e-waste management activities, supporting the government to have better control of the e-waste material flow [3,13].

Along with governmental actions, market-based sectors also contribute greater efforts to assisting and boosting households to recycle. Pioneer electronic producers consisting of HP, Apple, and Microsoft have initiated the annual program named Vietnamese Recycling Platform to build up waste collection and recycling manner, objecting at enhancing people's awareness and knowledge about recycling activities [34].

However, generally, it is apparent that the law and regulations and other recycling programs are still not well-managed and have not met the needs for e-waste management in underdeveloped countries yet [13]. Although the Vietnamese Government currently promulgates a Decree No. 155/2016/ND-CP to provide for penalties for administrative violations against regulations on environmental protection, this decree just focuses on violations against regulations on waste management in general, not specifically emphasizing e-waste. Therefore, it is difficult to control and penalize any organizational and individual entities who discard or treat e-waste incorrectly. Vietnam now confronts real difficulties in e-waste management, for instance, the insufficiency of specific e-waste legislation, the predominance of informal sectors (collecting and treating used EEE improperly) and the lack of statistical data [4,20]. It is reported that there is still a high quantity of e-waste treated under improper methods by informal collectors in several nations, especially in Asia and Africa, such as China, Bangladesh, India, Thailand, Vietnam, Nigeria, and Ghana [35]. These informal recycling activities not only cause serious threats to the environment but also induce a significant waste of valuable materials. The currently unsustainable e-waste management in Vietnam, plus to the worse e-waste trend force Vietnamese government soon to make a decision whether to establish an appropriate management system that can fulfill the global or regional e-waste standards [20].

Beside the burdens and challenges put on governmental organizations like policy-makers and authorities, the public knowledge and awareness about how to discard used appliances properly and environmentally friendly are also the most important factors which in turn help to control the flow of e-waste [3]. One of the main difficulties in winning the sustainability of e-waste management strategies is to create the link between formal and informal sectors and process and to provide residents with a better understanding about the risks of e-waste on human health for residents. As reported, several schemes and policy options are achieving desultory considerable effects; however, there is a need to make these environmental policies consolidated, which requires the integration among stakeholders' responsibilities [13].

The future for the accomplishment of sustainable e-waste management approach encompasses both challenges and opportunities for relevant stakeholders including consumers, entrepreneurs,

governments (authorities, regulatory agencies) and non-governmental organizations as well. The inter-cooperation and goal-oriented actions on e-waste are what is needed to gain the synthetic strength for running the e-waste management scheme smoothly [36].

### 2.2. Theory of Planned Behavior

The TPB was extended from the theory of reasoned action (TRA) by adding perceived behavioral control constructs, is considered one of the most effective socio-psychological models in predicting and explaining social behaviors [37–39]. In TPB, the performance of individuals' behavior is determined by their behavior intentions which influenced by three conceptually independent constructs, consisting of attitudes toward behavior (personal attitude and individual conduct), subjective norm (influence or social pressures to perform a behavior), and perceived behavioral control (indicates individuals' perceived ease or difficulty of performing the particular behavior) [40,41].

As of its launch, the TPB has been used to explore a wide variety of sustainable behaviors and has gained considerable success. For instance, it has been applied to sustainable transportation use [42], sustainable consumption [43], household energy-saving [44], personal standpoint on sustainable development [45], and waste management and composting [46]. The TPB has recently been widely applied as a theoretical framework in researches on the determinants of e-waste recycling behaviors and intentions. It is apparent that TPB is a useful and powerful framework for supporting researches related to sustainable behavior and this approach can be applied to most behaviors [47]. To be specific, there is a strong evidence proving that TPB is successful in most research on recycling [23,48]; therefore, TPB is now considered as a preferred theory providing a systematical framework for analyzing the determinant elements affecting recycling behavior [23,48].

Recently, researchers have extended the theoretical model by adding past recycling experience (among other variables) to the TPB in several studies about recycling behavior, with the hope to improve the predictability of the model. The reason behind this extension is that past recycling experience is supposed to have a considerable contribution to sustainable behavior. In fact, Ajzen [40] himself claimed: "The theory of planned behavior is, in principle, open to the inclusion of additional predictors if it can be shown that they capture a significant proportion of the variance in intention or behavior after the current variables of the theory have been taken into account". For example, among others, the exclusion of past behavior was detected as one of the limitations of the TPB [49].

The present study, therefore, will make an effort to test the tendency of all of these factors to elicit individuals' recycling intentions and behavior in a Vietnam, a developing country with very low e-waste recycling rate. It is expected to provide references and sources of information for the future studies, especially a source of the theoretical concept of influencing divers for residents' behavior and attitudes towards e-waste recycling.

### 2.3. Factors Affecting Individuals' Recycling Behavior

The behavioral intention of residents to take part in e-waste recycling programs is one of the primary elements in the plan of an e-waste management system. This leads to the fact that the success of a recycling program heavily relies on the engagement of residents. Hence, it is essential to understand the public perceptions on e-waste recycling to design effective policies which can help to deal with e-waste problems. Hornik et al. [50] classified the determinants that motivate consumers' behavior recycling under four main categories, including intrinsic incentives, extrinsic incentives, internal facilitators, and external facilitators. Among these variables from four listed groups, internal facilitators were ranked the strongest antecedents affecting recycling behavior of consumers with two main internal predictors of a predisposition to recycle were consumer knowledge and commitment to recycling. Monetary rewards and social influence were two variables among the group of external incentives, which came behind in terms of the heaviest influential antecedents [50].

Several studies have recently been conducted to find out the factors affecting the consumers' attitude and behavior toward e-waste recycling. Take a case study of e-waste recycling in China as an

example, Yin et al. [25] reported that only 47.9% of 1035 interviewees expressed their willingness to pay for used cell phone recycling with the fee mostly stretching from 0 to 5% of the recycling costs, compared to 1% advance recycling fee (ARF) which most California respondents were willing to pay for an e-waste recycling program [51]. This research also pointed out that region, education level, and average income significantly influenced the consumers' behavior [25]. In agreement with Yin et al. [25], Wang et al. [8] also found that income and costs of recycling were one of the main determinants affecting residents' behavior intentions towards e-waste recycling, beside environmental awareness, attitude towards recycling, perceptions of informal recycling, and norms and publicity [8]. In another study conducted in China as well, along with residential conditions and economic benefits, recycling habit and convenience of recycling facilities and service were also two key additional antecedents of willingness and behavior of Beijing residents in e-waste recycling [24]. In addition, there were several references have shown that information and environmental awareness motivate residents to participate in recycling [31,48]. It was also found that age, educational level, and monthly income were supportive factors influencing residents' behavior to participate in e-waste recycling [52]. Adding one more factor, that is household size, Sidique et al. [22] concluded that these demographic factors all influenced the recycling behavior of their respondents.

## 3. Theoretical Framework and Hypotheses Development

As discussed in the literature review section, TPB postulates three pillars which affect behavioral intention, consisting of attitude toward behavior, subjective norm, and perceived behavioral control [40]. In this study, to investigate the antecedents of e-waste recycling behavioral intention, the expansion of TPB has been employed as a research framework which is demonstrated in Figure 1. The recycling behavioral intention in this paper is defined as residents' likelihood and willingness to recycle e-waste to formal recycling sectors in the future.

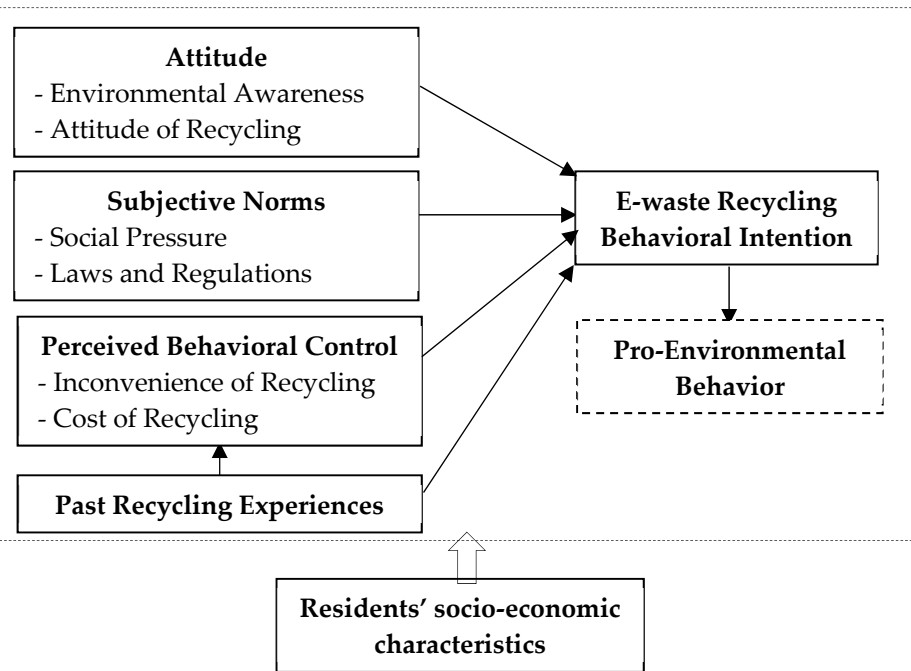

**Figure 1.** Conceptual framework.

The attitude in the conceptual framework is defined as residents' environmental awareness and attitude toward e-waste recycling. These two variables have been proved to have a significant impact on recycling behavioral intention [8,23,53]. The second pillar in the framework is subjective norm, which embodies to social pressure and laws and regulations. Hick et al. [18] and Wang et al. [8] revealed

that government laws and regulations play as a crucial factor influencing residents' recycling intention in the case of China, a country whose e-waste situation and government system are similar to those in Vietnam. In agreement with this concept, recent studies about e-waste management in Vietnam also emphasized the role of formal organizations in encouraging people to join in e-waste recycling program [13,20]. Along with laws and regulations, social pressure which has an impact on individuals' perception was considered an important factor in research of recycling behavioral intention [29,46]. The third pillar mentioned in the framework is perceived behavioral control which refers to the inconvenience of recycling and the cost of recycling. It is noticed that two factors presented to perceived behavioral control in this study come from residents' perception whether they are provided a good condition to join in the e-waste recycling program. Cost of recycling and inconvenience of recycling have been considered as important measures for recycling intention [8,23,34].

Besides, past experience was entered in the conceptual framework with the aim to elicit the impact of this variable on residents' behavioral intention. Past experience in this study means as an action and experience of residents toward recycling which they have been familiar with. Earlier studies have reported that once individuals have past experience in carrying out waste recycling, they show greater intention to perform it regularly [54]. That is the reason why past experience was added to this research framework.

In addition, demographic information—including age, gender, education, income, residential area—was included in this framework, with the purpose of detecting the effects of socio-economic characteristics on residents' e-waste recycling intention. When looking at the willingness to recycle, specifically in the case of e-waste, demographic variables such as age and education were proven to have the largest impacts. Moreover, gender and residential area also had a significant influence on recycling behavior [21].

From the theoretical model, the previous related studies, and the actual situation of e-waste in Vietnam, a research hypothesis model adopted in this study is given in Figure 2.

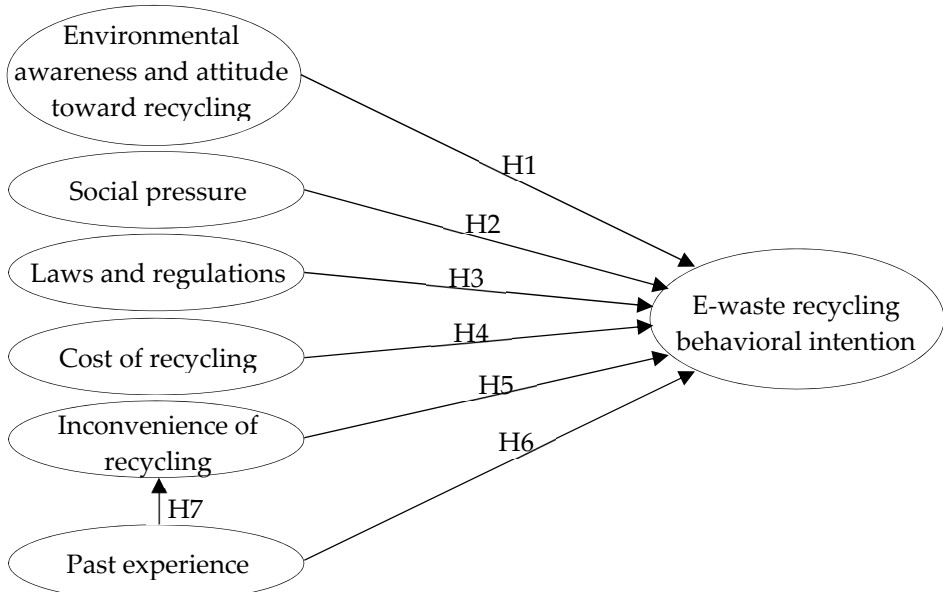

**Figure 2.** Hypotheses framework.

Tonglet et al. [23] and Nixon and Saphores [51] believed that attitude toward recycling and the environmental protection consciousness effectively stimulated the e-waste recycling behavioral intention of residents. In addition, Hansmann et al. [55] found that Switzerland citizens' knowledge and attitude toward recycling had an actively positive impact on their battery recycling. Therefore, the  following hypothesis is proposed:

**Hypothesis 1 (H1).** *Environmental awareness and attitude toward recycling positively affect residents' e-waste recycling behavioral intention.*

Oskamp et al. [56] revealed the most impressive outcome of their study was the positive impact of friends, family, and neighbors performed as a social influence of society on the subjects' participant in recycling. In fact, social pressure was found to be a motivational factor for recycling which effectively ensured the engagement of large communities in recycling. Similarly, the conclusion from the research conducted by Barr [57] showed that the correlations between recycling behavior and subjective norms were statistically meaningful. With regard to this concept, the following hypothesis is proposed:

**Hypothesis 2 (H2).** *Social pressure positively affects e-waste recycling intention.*

Yu et al. [58] proved that the laws and regulations had a positive effect on the willingness of residents to recycle e-waste. Wang et al. [8] found that promulgation and public spread of the laws and regulations improved environmental awareness among residents and in turn making them ready to recycle e-waste. In a nutshell, laws and regulations ruled by the government clearly play a vital part in e-waste recycling. Accordingly, the following research hypothesis can be concluded:

**Hypothesis 3 (H3).** *Laws and regulations positively impact e-waste recycling behavioral intention.*

It is believed that the more costly the expenditure on e-waste recycling is, the weaker likelihood of residents' intention is. According to Wang et al. [8], the residents' intention toward recycling of e-waste decreased when the cost of recycling increased, even residents might refuse to participate in formal recycling program if they had to pay more. However, other researches also concluded that recycling cost was not statistically significant [31]. Based on these references, the following research hypothesis is proposed:

**Hypothesis 4 (H4).** *Costs of recycling have an impact on e-waste recycling behavioral intention.*

Saphores et al. [31] revealed that recycling convenience, prior e-waste recycling experience was one of the most crucial internal variables for interpreting residents' willingness toward e-waste recycling. In fact, several studies have reported that perceived inconvenience had a negative impact on the likelihood of residents to participate in recycling activities [59–61]. Also, Wang et al. [24] recently explored Beijing residents' attitude toward e-waste recycling and their results revealed that recycling convenience and recycling habits had a meaningful explanation for willingness to recycle e-waste. Past behavior had a tendency to foresee intentions and future behavior [62–65]. Moreover, in the research on domestic solid waste in Glasgow, Knussen et al. [30] indicated there was a significant independent relationship between past recycling behavior and population's recycling. From these views, two other hypotheses are proposed:

**Hypothesis 5 (H5).** *The inconvenience of recycling has a negative significant impact on recycling behavior.*

**Hypothesis 6 (H6).** *Past experience of residents positively influences recycling intentions to e-waste.*

While several studies reported that past experience directly impacted on recycling intention instead of being a mediator in the model [63]. Ajzen [40] claimed that it is a behavioral experience that is one of the important elements which contributes to the formation of attitude, subjective norm and perceived control. In this study, the finding from Ajzen [40] was applied to test a hypothesis:

**Hypothesis 7 (H7).** *ast experience of residents negatively influences inconvenience of recycling.*

## 4. Research Methodology

### 4.1. Questionnaire Design

The questionnaire was built up based on the literature review, experts' opinions through in-depth interviews, combined with self-developed measurements. Six kinds of common e-waste—namely refrigerators, air conditioners, washing machines, computers-PCs (desktop PCs and laptop PCs), TV sets, and mobile phones—as research target objects. The questionnaire in this study includes three main sections, which was designed to fulfill the research objectives and a number of key requirements from research hypotheses.

Beginning with the warm-up section, some basic questions about respondents' knowledge of e-waste was designed in order to have a general picture about the understanding of residents toward e-waste. According to the answers received from this section, the researcher could exclude some questionnaires being detected that these interviewees had no idea about e-waste, which would affect the quality of later questions in the main section.

The second section focuses on the measurement of the constructs in the research model. Questions on environmental awareness and attitude towards recycling, social pressure, laws and regulations, cost of recycling, inconvenience of recycling, past experience and behavioral intentions were included. Minimum three items for each construct were set and a five-point Likert scale was used to weight each construct from different dimensions. The scale was ranged from 1 for "strongly disagree" to 5 for "strongly agree". To measure item "During the past three months how frequency did you recycle your waste at home" belonging to the construct of past experience, the scale was anchored at 1 for "never" to 5 for "always". The summary of constructs and their items is presented in Table 1.

In the third section, questions on demographic characteristics were asked, including gender, age, education level, monthly income, and residential area. With personal information collected from this section, the question of whether socio-economic variables have impacts on residents' recycling intention was explored.

**Table 1.** Constructs and measurements

| Constructs | Items | Measurements | References |
|---|---|---|---|
| Environmental awareness and attitude towards recycling (AAR), 7 items | EA1 | E-waste recycling is the main way to reduce the use of landfills and emissions of greenhouse gasses. | [8,22,30,46,53,66] |
| | EA2 | E-waste recycling is a primary way to conserve natural resources. | |
| | EA3 | E-waste recycling improves the quality of the environment. | |
| | AR1 | I feel very satisfied when recycling e-waste. | |
| | AR2 | E-waste recycling is useful to create a better community environment. | |
| | AR3 | E-waste recycling is everyone's responsibility to reduce the volume of e-waste generated. | |
| | AR4 | I am not interested in the idea of e-waste recycling. | |
| Social Pressure (SP), 3 items | SP1 | If my family and friends are involved in e-waste recycling, I will also engage in it. | [8,53,54] |
| | SP2 | The media influences me to e-waste recycling. | |
| | SP3 | The community where I live would influence me to participate in recycling e-waste. | |
| Laws and regulations (LR), 3 items | LR1 | Vietnamese laws well require the responsibilities of residents to recycle e-waste. | [8,66] and self-developed for LR1 and LR3 |
| | LR2 | Government policy would influence me to recycle e-waste. | |
| | LR3 | If there are laws and or regulations related to e-waste recycling, I will obey them. | |
| Cost of recycling (CR), 3 items | CR1 | Recycling programs are costly. | [8,23] |
| | CR2 | I think expenditure on transportation of e-waste to the recycling center is high. | |
| | CR3 | I think handling charges of e-waste recycling are high. | |
| Inconvenience of Recycling (ICR), 4 items | ICR1 | I feel difficult to sort e-waste for recycling. | [8,22,23,53] |
| | ICR2 | I have no time to send e-waste to the collection point. | |
| | ICR3 | It is inconvenient to transport e-waste to the collection point. | |
| | ICR4 | I think neighboring e-waste recycling channels are deficient. | |
| Past experience (PE), 3 items | PE1 | I am well acquainted with the recycling facilities. | [22,24,67] |
| | PE2 | I am knowledgeable about the materials suitable for recycling. | |
| | PE3 | During the past three months how frequently did you recycle your waste at home. | |
| Behavioral Intention (BI), 4 items | BI1 | I am willing to contact formal e-waste recycling organizations to deal with e-waste in the future. | [8,66] and self-developed for BI2 and BI4 |
| | BI2 | I intend to drop-off my e-waste if formal collection systems are available. | |
| | BI3 | I am willing to participate in environmental programs hold by the government. | |
| | BI4 | I am willing to tell my relatives about the e-waste recycling experiences. | |

### 4.2. Pilot Test

A pilot study on 50 respondents was carried out with the purpose of ensuring the reliability and validity of the data collection tool as well as detecting the potential problems related to questionnaires. Each interviewee was invited to fill in all questions in the survey with the presence of interviewer and then their feedbacks and opinions were collected and analyzed in order to correct the questionnaire. Cronbach's alpha ($\alpha$) was used to test the internal consistency of the variables which refers to the reliability of the data. In the pre-test, all items in the questionnaire were passed the threshold of Cronbach's $\alpha$ which was set at greater than 0.7, meaning high reliability [68]. In addition, the clarity of all questions in the questionnaire was diagnosed by asking these participants how easy they understand the questions. Consequently, minor changes in wording were made and some questions were rephrased.

### 4.3. Survey Design

This research, with the aim to focus on the residents living in urban areas, conducted the survey in Danang city including six administrative divisions: Thanh Khe, Hai Chau, Son Tra, Ngu Hanh Son, Lien Chieu, and Cam Le, with 197,987 households in total (data collected from Appendixes in Resolution 108/NQ-HDND promulgated by Danang people's councils in the year 2017). There are several reasons for selecting Danang as a location of this survey. The first and most important reason is that Danang has recently experienced an extreme growth of population and a rapid development in terms of ICT with a huge rise in electrical and electronic consumption. Thus, consideration about e-waste problem is now very important to prevent this emerging energetic city from becoming an e-waste dumping area. In addition, Danang is the primary and largest port of the central area of Vietnam; hence, the problem of e-waste illegal boundary transportation also raises a concern among e-waste usage, recycling and e-waste management. Last but not least, there is currently no e-waste recycling program implemented in Danang, while a pilot project to collect and treat e-waste, called "Vietnam Recycles" has recently launched in three other largest cities namely, Hanoi, Ho Chi Minh City, and Haiphong. Therefore, it is hoped that this research will help to understand residents' attitude and behavioral intention, which is useful for implementing the policy of e-waste recycling not only in Danang, but also in all cities in Vietnam. This is to collect and process all used or defective electronic products in a safe and professional manner to ensure highly professional waste treatment and achieve maximum recovery of natural resources.

In this study, householders living in six above listed administrative divisions were selected to be the target respondent of this research. There are two major reasons for this selection. Firstly, the main source of e-waste generation originated from each household. Secondly, since EEE is considered to be a common property sharing by all members of a household, the disposal of used EEE is a family practice.

Data for this research were collected using a face-to-face survey through the distribution of 545 questionnaires to available Danang households during two months from July to August of 2018. The survey process was under the concept of stratified random sampling which involved dividing the entire population by districts which were called strata. Proportional allocation to each stratum based on its size relative to the population and then the samples were drawn randomly according to the proportion in each district under the principle of systematic random sampling, which made the collected data trustworthy. The respondents were chosen based on the principle of systematic random sampling without consideration of any special criteria. The first interviewee was first randomly selected from the population and then, each nth interviewee in the list was selected, where n is the sampling interval which was calculated by dividing the total number of the population with the sample size.

After screening and removing invalid questionnaires (those were not fully completed or the answers were one-scale dimension), a total of 520 valid samples were used for further analysis, presenting an effective recovery rate of 95.41%. Table 2 presents the detail information of respondents' demographic characteristics.

**Table 2.** Demographic profile of respondents (*n* = 520)

| Demographic Variables | Group | Frequency | Percentage (%) |
|---|---|---|---|
| Gender | Female | 309 | 59.4 |
| | Male | 211 | 40.6 |
| Age | ≤20 | 41 | 7.9 |
| | 21–30 | 146 | 28.1 |
| | 31–40 | 173 | 33.3 |
| | 41–50 | 84 | 16.2 |
| | 51–60 | 46 | 8.8 |
| | >60 | 30 | 5.8 |
| Educational level | Lower secondary | 12 | 2.3 |
| | Upper secondary | 69 | 13.3 |
| | College/Vocational education | 69 | 13.3 |
| | University | 283 | 54.4 |
| | Master's degree or above | 87 | 16.7 |
| Residential Area | Thanh Khe | 161 | 31.0 |
| | Hai Chau | 106 | 20.4 |
| | Lien Chieu | 70 | 13.5 |
| | Son Tra | 77 | 14.8 |
| | Ngu Hanh Son | 44 | 8.5 |
| | Cam Le | 62 | 11.9 |

*4.4. Data Analysis*

Two main analysis steps consisting of the exploratory factor analysis (EFA) and SEM were used in this study. Firstly, the EFA was adapted to explore the underlying construct of a set of items. Then, SEM was applied with the aim of analyzing the structural relationship and in turn finding out the factors affecting residents' e-waste recycling behavioral intention.

The dataset was initially experiencing preliminary analysis to test the initial scale. By applying the statistical package for social sciences (SPSS 22.0) software, EFA was performed to examine the strength and relationship between each common factor to the corresponding measure. The value of KMO (Kaiser–Meyer–Olkin) was 0.893 which was greater than the critical value of 0.7 [68], and the significant value in the Barlett sphericity test was *p* = 0.000, which indicated that the original dataset was appropriate for factor analysis. In the step of EFA, the principal component analysis with varimax rotation and factor loadings which were greater than 0.5 was used to extract constructs from all original items. The results of EFA, as shown in Table 3, presented that there were six factors extracted with explained variance was 70.199%. It was noted that almost all items remain the same as those in initial scale from hypotheses framework, except for two items CR1 and ICR4 whose factor loadings were lower than 0.5, and then were deleted.

**Table 3.** Varimax-rotated component analysis factor matrix and Cronbach's $\alpha$ values

| Constructs | Items | Main Factors | | | | | | Cronbach's $\alpha$ |
| --- | --- | --- | --- | --- | --- | --- | --- | --- |
| | | 1 | 2 | 3 | 4 | 5 | 6 | |
| Environmental awareness and attitude toward recycling (AAR) | AR2 | 0.856 | | | | | | 0.904 |
| | EA1 | 0.817 | | | | | | |
| | EA3 | 0.797 | | | | | | |
| | EA2 | 0.793 | | | | | | |
| | AR3 | 0.739 | | | | | | |
| | AR1 | 0.727 | | | | | | |
| | AR4 | 0.584 | | | | | | |
| Laws and regulations (LR) | LR1 | | 0.799 | | | | | 0.821 |
| | LR2 | | 0.781 | | | | | |
| | LR3 | | 0.678 | | | | | |
| Inconvenience of recycling (ICR) | ICR2 | | | 0.866 | | | | 0.777 |
| | ICR3 | | | 0.854 | | | | |
| | ICR1 | | | 0.754 | | | | |
| Social pressure (SP) | SP2 | | | | 0.812 | | | 0.848 |
| | SP3 | | | | 0.696 | | | |
| | SP1 | | | | 0.645 | | | |
| Past experience (PE) | PE1 | | | | | 0.825 | | 0.672 |
| | PE2 | | | | | 0.783 | | |
| | PE3 | | | | | 0.706 | | |
| Cost of recycling (CR) | CR2 | | | | | | 0.890 | 0.749 |
| | CR3 | | | | | | 0.859 | |

Extraction Method: principal component analysis. Rotation Method: Varimax with Kaiser normalization. Factor loading less than 0.50 have not been printed.

To assess the reliability of the revised scale, Cronbach's $\alpha$ values were calculated for internal validity. The result from Table 3 indicated that the Cronbach's $\alpha$ values of all variables were approximately or greater than 0.7, ranging from 0.672 to 0.904. Hair [68] suggested that Cronbach's $\alpha$ value should be higher than 0.700, but 0.600 is acceptable. Compared to the critical value of Cronbach's $\alpha$ suggested by Hair [68], it can be concluded that all variables in modified scale were internally consistent and reliable to conduct in this study.

Subsequently, SEM was adopted to test the proposed conceptual model by using analysis of moment structures (AMOS 22.0) software. Firstly, confirmatory factor analysis (CFA) was assessed to measure for the reliability, convergent and divergent validity, and then the structural model was tested to examine the structural path estimates and hypothesized relationships among constructs.

Finally, SPSS 22.0 software (IBM Corp, Armonk, NY, USA) was once again used to perform Independent *t*-test and one-way analysis of variance (ANOVA) for determining the impacts of demographic variables on all constructs.

## 5. Results and Discussion

This paper adapted the two-step analytical procedure to explain the results. Initially, the measurement model was tested by adopting CFA. The structural model was later assessed to test research hypotheses.

### 5.1. Measurement Model

Composite reliability, convergent validity, discriminant validity, and content validity of each variable were evaluated in this step by using CFA. As can be seen from Table 4, all the standardized factor loadings were significant with the value ranged from 0.501 to 0.960, above the recommended lower limit value of 0.5 [68]. The composite reliability for all constructs was higher than the cut-off value of 0.6, indicating good internal consistency for all latent constructs [68,69]. The average variance extracted (AVE) was greater than 0.5 for all constructs, presenting good convergent validity, except

for past experience (PE). Based on Fornell and Larcker [70], AVE of a construct can be accepted at the value of 0.4 in case that its composite reliability is greater than 0.6, the convergent validity of the construct is still adequate, thus, the value of 0.441 for a construct of PE can be accepted.

**Table 4.** Composite reliability and convergent validity of constructs and factor loadings of items

| Constructs | Items | Factor Loadings | Average Variance Extracted (AVE) | Composite Reliability (CR) |
|---|---|---|---|---|
| Environmental awareness and attitude toward recycling (AAR) | EA1 | 0.792 | 0.581 | 0.906 |
| | EA2 | 0.738 | | |
| | EA3 | 0.758 | | |
| | AR1 | 0.726 | | |
| | AR2 | 0.880 | | |
| | AR3 | 0.786 | | |
| | AR4 | 0.634 | | |
| Social pressure (SP) | SP1 | 0.866 | 0.653 | 0.849 |
| | SP2 | 0.735 | | |
| | SP3 | 0.817 | | |
| Laws and regulations (LR) | LR1 | 0.771 | 0.607 | 0.822 |
| | LR2 | 0.753 | | |
| | LR3 | 0.812 | | |
| Cost of recycling (CR) | CR2 | 0.625 | 0.656 | 0.785 |
| | CR3 | 0.960 | | |
| Inconvenience of recycling (ICR) | ICR1 | 0.610 | 0.55 | 0.783 |
| | ICR2 | 0.801 | | |
| | ICR3 | 0.797 | | |
| Past experience (PE) | PE1 | 0.795 | 0.441 | 0.696 |
| | PE2 | 0.664 | | |
| | PE3 | 0.501 | | |
| Behavioral intention (BI) | BI1 | 0.799 | 0.695 | 0.901 |
| | BI2 | 0.773 | | |
| | BI3 | 0.870 | | |
| | BI4 | 0.887 | | |

To test the discriminant validity of the constructs, the square root of AVE and the correlations among constructs were compared. In this study, discriminant validity was achieved because the square root of AVE of all constructs was greater than the inter-construct correlation values [70]. It is also suggested that the correlation estimates between constructs in CFA model must be not greater than 0.85 as a requirement for discriminant validity [71]. The output of AMOS for this study showed that all correlation estimates were lower than 0.85, illustrating that all constructs were unique. Content validity of this questionnaire was also ensured because the accomplishment of this questionnaire was made by the references of previous studies and actual situation, combined with the pieces of advice from experts, making that all items are representative of the outcome and questions cover the full range of the issues or problems being measured.

The overall fitness of measurement model including absolute, incremental and parsimonious indices of fit was examined by using the maximum likelihood (ML) method in AMOS. Table 5 shows the CFA results for measurement model fit indicators after identification, fitting, evaluating and modifying. As can be seen that the overall $\chi^2$ for this model was significant with *p*-value < 0.05, indicating that the fit of the data to the hypothesized model was not entirely adequate. This can be explained by the effect of large samples. The reason behind this problem is that $\chi^2$ is disturbed by the estimated parameters and sensitive to sample size. Consequently, $\chi^2$ statistic becomes to be unrealistic in most SEM empirical model because of its limitation [69].

**Table 5.** Goodness-of-fit test results for the measurement model

|  | Indicators | Criterion | Results | Judgment | References |
|---|---|---|---|---|---|
| Absolute fit measures | $\chi^2$ | $p > 0.05$ |  | Sensitive to sample size >200 | |
|  | GFI | > 0.9 | 0.933 | Good fit | |
|  | AGFI | > 0.9 | 0.913 | Good fit | |
|  | RMR | < 0.08 | 0.034 | Good fit | |
|  | SRMR | < 0.05 | 0.037 | Good fit | |
|  | RMSEA | < 0.08 | 0.041 | Good fit | |
| Incremental fit measures | NFI | > 0.9 | 0.934 | Good fit | [69,71,72] |
|  | TLI | > 0.9 | 0.961 | Good fit | |
|  | CFI | > 0.9 | 0.968 | Good fit | |
|  | IFI | > 0.9 | 0.968 | Good fit | |
|  | RFI | > 0.9 | 0.92 | Good fit | |
| Parsimonious fit measures | PGFI | > 0.5 | 0.718 | Good fit | |
|  | PNFI | > 0.5 | 0.778 | Good fit | |
|  | PCFI | > 0.5 | 0.807 | Good fit | |
|  | CMIN/DF | < 3 | 1.865 | Good fit | |

To address χ2 constraints, the goodness-of-fit indices were used to evaluate the fit of the model. One of the first fit statistics can be considered is the ratio of χ2 to the degree of freedom (CMIN/DF), which had the value of 1.865 (less than the recommended value of 3.000) in this research model, indicating a good adaptation of the model to the observed sample data. In addition, other most important indicators can be used to assess of fitness of the model include the goodness of fit index (GFI), the Tucker–Lewis index (TLI), the comparative fit index (CFI), the normed fit index (NFI), the root mean square error of approximation (RMSEA), the standardized RMR (SRMR), and the parsimonious normed fit index (PNFI). Furthermore, the other fit index values such as the adjusted goodness of fit index (AGFI), the root mean square residual (RMR), the incremental fit index (IFI), the relative fit index (RFI), the parsimony goodness-of-fit index (PGFI), the parsimonious normed fit index (PNFI), and the parsimonious comparative fit index (PCFI) were examined. The values of these indices from the research model and their threshold are both shown in Table 5 and it can be concluded that the measurement model had a good fit.

*5.2. Structural Model*

The structural model was performed to test research hypotheses. Table 6 shows the outcomes of the structural model with standardized path coefficients. The factors that had significant impacts on recycling behavioral intention were environmental awareness and attitude towards recycling, social pressure, laws and regulations, cost of recycling, and inconvenience of recycling. These constructs accounted for 56% of the variance in behavioral intention, illustrating that a substantial amount of variance can be explained by the constructs. Meanwhile, the variable of laws and regulations was the strongest predictor of individuals' intention. Although inconvenience of recycling and past experiences negatively affected residents' recycling intention, only the former one was statistically significant. In contrast, past experience showed the statistically significant negative effect on inconvenience of recycling. The significant relationship between all above factors in both positive and negative effects indicates that H1, H2, H3, H4, H5, and H7 was supported, while H6 was not verified because there was no statistically significant relationship between past experience and behavioral intention. Finally, the causal effects among seven latent constructs which consist of both direct and indirect effects are shown in Figure 3.

**Table 6.** Hypothesis test results—structural paths

| | Hypotheses | Structural Path Coefficients | *p*-Value | *t*-Stats | Comments |
|---|---|---|---|---|---|
| H1: | AAR —> BI | 0.259 | *** | 3.927 | Supported |
| H2: | SP —> BI | 0.137 | 0.049 | 1.967 | Supported |
| H3: | LR —> BI | 0.359 | *** | 5.488 | Supported |
| H4: | CR —> BI | 0.125 | 0.002 | 3.068 | Supported |
| H5: | ICR —> BI | −0.085 | 0.02 | −2.335 | Supported |
| H6: | PE —> BI | −0.031 | 0.443 | −0.768 | Not Supported |
| H7: | PE —> ICR | −0.101 | 0.078 | −1.763 | Supported |

Note: *** $p < 0.001$.

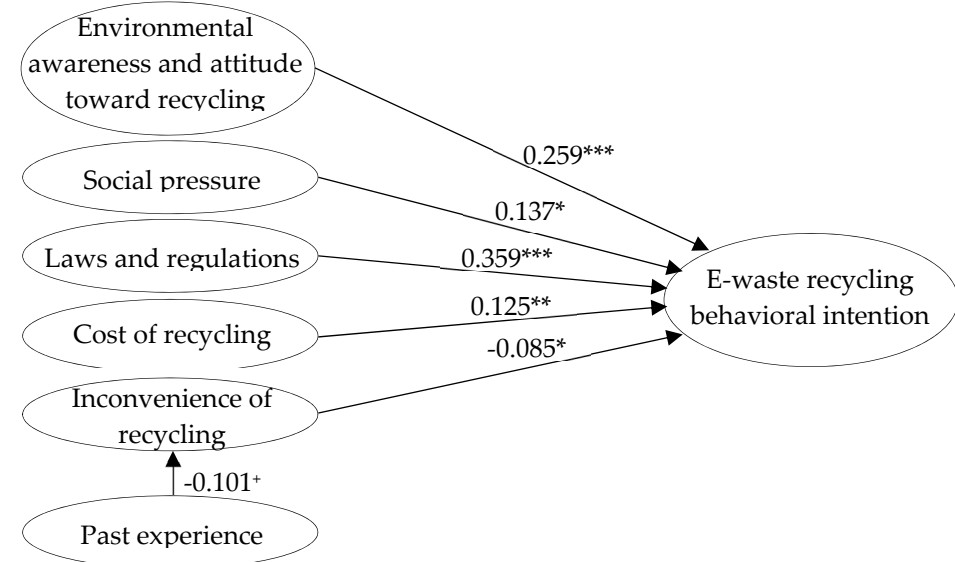

**Figure 3.** The estimated structural model. (Note: $^+$ $p < 0.10$, * $p < 0.05$, ** $p < 0.01$, *** $p < 0.001$).

The results of the SEM analysis indicate that laws and regulations, with the influence coefficient of 0.359, plays the most significant role in residents' e-waste recycling intention. This outcome was consistent with previous studies conducted by Wang et al. [8] and Nduneseokwu et al. [66]. These authors reported that laws and regulations positively affected citizens' participation in e-waste recycling programs. Moreover, connecting with the actual situation in Vietnam, a socialist country ruled by law and of the people, by the people, and for the people; hence, it is apprehensible that the legal system is one of the most important factors affecting residents' behavior. From this result, the vital role of laws and regulations is emphasized; therefore, there is a high-priority need for the introduction of enforced legislation, clearly defining the responsibility of residents in e-waste recycling.

The influence path coefficient of environmental awareness and attitude towards recycling was 0.259 with *p*-value reached a significant level (<0.001), representing that this construct had a significant positive effect on e-waste recycling behavior, being consistent with earlier studies [8,46,53]. This influence path coefficient came into the second place, behind that of laws and regulations, showing that environmental awareness and attitude towards recycling was the second strongest determinant factor constrained the residents' recycling intention effectively. It can be concluded that environmental awareness and attitude towards recycling which refers to the residents' awareness about environmental consequences of e-waste and responsibility toward environmental protection had a strong contribution, leading the development of people's favor and satisfaction to participate in recycling. In fact, by recognizing the harmful effects of e-waste on human health and the environment

if it is treated improperly, residents can raise more awareness and attitude toward recycling, and as a result, their recycling intention will grow.

With regard to the relationship between social pressure and e-waste recycling intention, it can be seen from the SEM analysis results that social pressure had a positive impact on e-waste recycling behavioral intention, similar to what was reported in previous studies [23,53,54]. Although this impact was significant at 5% level, the standardized coefficient was not very large, suggesting that the influence of social pressure was weaker than that of laws and regulations and environmental awareness and attitude towards recycling. This implies that residents believe that their families', friends', and neighbors' perception of e-waste recycling and the media would only influence to a certain extent to their e-waste recycling intention.

The results also revealed that the costs of recycling statistically affected e-waste recycling behavioral intention. Interestingly, the path coefficient of recycling cost and behavioral intention was at 0.125, meaning that e-waste recycling intention was not strongly affected by the cost and this factor had a positive impact on recycling intention. It is in disagreement with the results from some of the previous studies. For example, Wang et al. [8] reported that the Chinese residents' intention toward recycling of e-waste decreased when the cost of recycling increased and the higher the cost of e-waste recycling led to the weaker likelihood of intention. However, in the case of Vietnam, it can be explained that people concern more about their health than their expenditure they have to spend on recycling e-waste. This idea is also supported by the strong relationship between the awareness and recycling intention as analyzed above. When citizens have a high awareness of the risks of toxins in e-waste, they tend to pay more attention to protecting their health and to be willing to join in e-waste recycling despite the cost they have to pay. In addition, residents in Vietnam, and Danang as well, are currently very active in paying a fee for solid waste treatment. Even, according to the research of Nguyen et al. [73], residents in Hanoi (Vietnam) showed their willingness to pay a cash fine for each time they destroy the rule of waste separation, this is evidence indicating that the cost is not a big concern of residents when they agree to participate in recycling activities.

Many researchers believe that convenience is one of the main factors positively influencing recycling intention. This factor is evaluated from the respondents' ability and access to e-waste collection or recycling centers surroundings their residential area. In other words, inconvenience may have negative impacts on residents' e-waste recycling intention [9,24,53]. In agreement with these studies, our study also shows the same results, which its coefficient was −0.085, referring that residents suppose that e-waste recycling is not the easy task but inconvenience plays a super weak impact on the behavioral intention of recycling.

Moving to the last construct, that is past experience, while several studies supported that past experience had a positive effect on recycling behavior [22,67], there are still some studies which revealed that this factor had no significant impact on recycling intention of residents. For this current research, it was also found that there was no direct relationship between past experience and e-waste recycling intention. However, there exists an indirect relationship among them. The construct played a mediator role in this research framework was inconvenience of recycling. In fact, past experience showed a negative influence on recycling inconvenience; however, this effect was weak, thus there is a need to do more research to elicit the relationship among these two factors.

*5.3. Contributions of Demographic Variables*

It is undeniable that the demographic variables have a great contribution to predicting residents' pro-environmental behavior [54]. Sidique et al. [22] summarized that socio-economic aspects including age, education level, and family size had an important impact on recycling attitude and behavior. However, Wang et al. [24] indicated that education, income, and the size of the household were not meaningful in giving an explanation for e-waste recycling willingness of residents. Some of the scientific reports suggested that seniors in society had more tendency on recycling [60,74–77]; in contrast, others revealed that the relationship between age and recycling was not significant [56,78–80].

Domina and Koch [81] claimed that this clear difference might root from a generational effect. In addition, Hansmann et al. [55] gave the further suggestion that discrepancies among different cultures of various countries may be an influencing reason. In terms of gender, women had much more readiness to recycle than men did [82,83]; however, along with Gamba and Oskamp [84], Werner et al. [80] and Pinto et al. [61] found that there was no statistically significant correlation between gender and recycling. From these concepts, gender, age, income, education level, and residential area (as five demographic variables) was taken into consideration in this study.

Independent *t*-test and one-way ANOVA were performed to examine whether these selected demographic variables play an important impact on determinants that have an influence on residents toward e-waste recycling. The statistic results of the relationship between socio-economic variables and e-waste recycling intention variables are depicted in Appendix A (Tables A1 and A2). From the results, it is apparent that gender, age, and education level had statistically significant impacts on environmental awareness and attitude toward recycling. Besides, the impact of laws and regulation on residents was affected by age and education level while residents' recycling behavioral intention was influenced by gender and education level. There was a significant relationship between age and inconvenience of recycling and past experience as well. Importantly, income and residential area were not statistically significant for any factor.

Subsequently, five demographic variables were selected to operate in SEM to explore the distinction of effects of socio-economic characteristics on e-waste recycling behavioral intention performance. The results collected after running AMOS 22 shows that only gender had a moderating effect on e-waste recycling intention, the other four variables—namely income, age, education level, and residential area—have no significant effect on residents' behavior intention. The outcome of standardized path coefficients for gender is represented in Table 7 as below.

**Table 7.** Standardized path coefficients for gender

| Hypotheses | | Path Coefficients | | Significance |
|---|---|---|---|---|
| | | **Male** | **Female** | |
| H1: | AAR —> BI | 0.246 * | 0.237 ** | Both |
| H2: | SP —> BI | 0.315 ** | 0.004 | Males |
| H3: | LR —> BI | 0.228 * | 0.494 *** | Both |
| H4: | CR —> BI | 0.096 | 0.164 ** | Females |
| H5: | ICR—> BI | −0.138 * | −0.042 | Males |
| H7: | PE—> ICR | −0.089 | −0.122 | No sig. |

Note: * $p < 0.05$, ** $p < 0.01$, *** $p < 0.001$.

The relationship between environmental awareness and attitude toward recycling and behavior intention (H1), and laws and regulations and behavior intention (H3) were both significant for two groups, male and female. On the other hand, the paths from social pressure and inconvenience of recycling to behavior intention (H2 and H5, respectively) were only significant for male group, while that from costs of recycling to intention (H4) was significant for females. Consequently, compared to females, males tend to be much influenced by friends, neighbors and family members on e-waste recycling intention. Besides, inconvenience of recycle was also a factor contributing less effect on recycling intention of men than that of women. In contrast, females showed a stronger motivational effect of recycling cost on e-waste recycling intention than that of males. From that, it can be concluded that males were the main targeting group that contributes to the success of e-waste recycling programs. Fortunately, in the context of Vietnam society which is affected by Confucianism, a male always plays the main role in each household so that if the man has a strong motivation toward e-waste recycling, that will help to make the e-waste program more effective.

## 6. Conclusions and Policy Implication

The findings from this study demonstrated various drivers of residents' e-waste recycling behavioral intention in which concerns about laws and regulations exert the strongest influence on e-waste recycling behavioral intention of both men and women. As this study shows, there is a top-priority need for introducing legislation solving e-waste problems and encouraging residents' engagement in e-waste recycling in Vietnam. Enforced legislation is one of the best examples of stricter measures which should be put in place to control the e-waste issue at the point sources, and in this way, it may lead to an increasing effectiveness of recycling rate. Laws and regulations which emphasize the incorporation of all relevant stakeholders' responsibilities should be established and implemented in order to contribute for a successful e-waste recycling and management system.

Besides, the findings from this study revealed that environmental awareness and attitude toward recycling attitude is the primary influencing factors in activating residents' e-waste recycling intention toward formal collections. That shows the fact that those who are participating in e-waste recycling do so mainly because they understand that such behavior contributes to save natural resources and eliminate the environmental problem. Therefore, it is indispensable to build up educational campaigns which may raise people's awareness and beliefs about the benefits of recycling in conserving natural resources as well as reducing the use of landfills and emissions of greenhouse gasses. By realizing how beneficial e-waste recycling is, it would motivate residents to recycle their discard e-products and further foster the e-waste recycling habits among residents.

In addition, with regard to the significance of social pressure in developing residents' e-waste recycling, public media and communication campaigns should be designed appropriately with the aims to attract more and more people to perform recycling behavior. Educational and communication strategies should be taken to equip residents with the knowledge about the proper methods to recycle and reuse e-waste for the residents. Especially, as can be seen from the results, males should be considered to be the main targeting group of such communication programs. Another important factor that helps to encourage residents to participate in formal e-waste collections is the convenience of recycling. In fact, the result of this study depicts that recycling behavior is impeded by the inconvenience of recycling. Thus, the need for providing the available accessibility of e-waste collection infrastructure may be put as one of the top priorities.

From the result indicating that cost of recycling is not a barrier to hamper residents to recycling e-waste, it provides a meaningful reference for law-makers who can follow the model of e-waste recycling in other countries to develop the effective laws and regulations solved e-waste problems in Vietnam. Take a case of Japan as an example, the e-waste management system is implemented under the Home Appliance Recycling Law, which clearly determines the roles and responsibilities of relevant stakeholders, including manufacturers, retailers, and consumers. Meanwhile, consumers have responsibilities for properly discharging used home appliances, not being allowed to sell e-waste to informal collectors and bearing costs for collection and recycling of used home appliances [85]. Such a Japanese e-waste management mechanism is worth-learning and it is hoped that the similar system for controlling e-waste will be soon applied in Vietnam.

From the findings from this study, linked with the current situation of e-waste in Vietnam, it can be suggested that the application of the taking-back system on e-waste management should be emphasized and improved by using specific law and regulation. The success of this system also requires the cooperation of all stakeholders, including governmental organizations, consumers, manufacturers, retailers, and non-governmental organizations. To be more specific, the government has an obligation to establish the law and regulation, supervise the implementation of organizational and individual entities, and penalize violated agencies. Consumers share the responsibility by bringing their discarded products back to formal collection centers. Manufacturers and retailers need to work together and better seek partnerships with third-party logistics providers which will help decrease e-waste through reverse logistics for used and obsoleted EEE. Reverse logistics in e-waste is also well-known for its efficient resource usage which involves valuable material recovery. This will help

reduce the cost of raw materials, enhance the cost-effectiveness of manufacturers, and eliminate the negative impact on the environment. Therefore, it is necessary for EEE manufacturers to develop their reverse logistics appropriately for profit-oriented purposes of gaining benefit and reducing cost.

In summary, this study builds upon theoretical lens of the TPB, a cognitive psychological model, providing a better understanding of how attitudes, subjective norms, and perceived behavioral control influence recycling behavior. The findings from this work showed the determinants of residents' behavioral intention to participate in a formal e-waste recycling system in Danang, one of the fastest growing startup cities in Southeast Asia. A thorough understanding of residents' intention toward e-waste recycling, therefore, makes a considerable contribution to the establishment and implementation of effective e-waste reuse and recycle management system.

It is obvious that the study of e-waste recycling behavioral intention is extremely important which is considered as a milestone and creates a solid foundation in paving the success of e-waste management in an emerging country like Vietnam. Especially in the circumstance of the rapidly increasing amount of e-waste, while the currently existing e-waste legislation is not truly effective. In addition, this study is among a very few and maybe the first one discussed in the area of sustainable behavioral intention toward e-waste recycling and pro-environmental behavior as well; thus its contribution to the theoretical and practical aspects of e-waste recycling field is brilliant, providing better suggestions and solutions for improving recycling systems. It is also a useful source of information for planning and improving international collaboration to resolve e-waste issues for the international community as a whole.

**Author Contributions:** H.T.T.N., R.-J.H., and C.-H.L. were responsible for the theoretical framework formulation, research design, data analysis, and writing of the manuscript. H.T.T.N. and H.T.T.N. managed with data collection.

**Funding:** This research received no external funding.

**Acknowledgments:** Authors would like to thank all respondents who spent their valuable time for answering questionnaires.

**Conflicts of Interest:** The authors declare no conflict of interest.

## Appendix A

**Table A1.** Analysis of variance of demographic characteristics

| Demographic Variables | Gender | | Age | | | | Education Level | | | |
|---|---|---|---|---|---|---|---|---|---|---|
| Constructs | F | *p*-Value | F | *p*-Value | Scheffe (a) | Tamaha (b) | F | *p*-Value | Scheffe (a) | Tamaha (b) |
| AAR | 3.499 | 0.011 | 3.583 | 0.003 | (3)>(2) | | 3.210 | 0.013 | (v)>(iv)>(ii) | |
| SP | 0.184 | 0.064 | 3.658 | 0.003 | - | | 0.872 | 0.480 | - | |
| LR | 0.490 | 0.068 | | | | (3)>(1) | 4.060 | 0.003 | (v)>(iv)>(ii) | |
| CR | 0.494 | 0.936 | 1.230 | 0.293 | - | | 0.980 | 0.418 | - | |
| ICR | 0.244 | 0.946 | | | | (6)>(3)>(2)>(1)>(5) | 0.777 | 0.541 | - | |
| PE | 4.626 | 0.075 | 3.768 | 0.002 | (3)>(1) | | | | | - |
| BI | 3.419 | 0.031 | 1.391 | 0.226 | - | | 3.007 | 0.018 | (v)>(ii) | |

**Table A2.** Analysis of variance of demographic characteristics (continued)

| Demographic Variables | Income | | | | Residential Area | | | |
|---|---|---|---|---|---|---|---|---|
| Constructs | F | *p*-Value | Scheffe (a) | Tamaha (b) | F | *p*-Value | Scheffe (a) | Tamaha (b) |
| AAR | | | | - | | | | - |
| SP | | | | - | 1.385 | 0.228 | - | |
| LR | | | | - | 0.644 | 0.666 | - | |
| CR | 1.162 | 0.327 | - | | 1.160 | 0.328 | - | |
| ICR | | | | - | 2.910 | 0.013 | - | |
| PE | 0.653 | 0.625 | - | | 0.325 | 0.898 | - | |
| BI | | | | - | 1.928 | 0.088 | - | |

Note: -: no significant mean difference. 1, 2, 3, 5, and 6: below 20, (21-30), (31-40), (51-60), and above 60 years old, respectively. ii, iv, and v: education level of upper secondary, university, and master's degree and above, respectively. (**a**): Scheffe test was used when significant *p*-value from Test of Homogeneity of Variances greater than 0.05. (**b**): Tamaha test was used as a post-hoc test when *p*-value from Test of Homogeneity of Variances lower than 0.05.

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
