# Peer review of "Determinants of Residents’ E-Waste Recycling Behavioral Intention: A Case Study from Vietnam"

_sustainability, doi:10.3390/su11010164_

Reviewer 1 Report

1. Reduce the introduction and bibliographic review by removing information that is already consolidated in the scientific community, leaving only the referenced data that is relevant to the article.

2. Improve the quality of the figures.

3. Replace tables by graphs; it is difficult to interpret the data as presented.

4. Correct all the apostrophes by following "s".

5. Remove the first person from the entire article.

6. It was not made clear in the article how the interviewees were chosen, according to the region, if she is poor, has municipal recycling policies and offers means to recycle e-waste.

7. It was not clear in the article whether there are penalties for the incorrect disposal of e-waste.

8. It is necessary to highlight in the article the economic factors involved to improve recycling of e-waste, for example, reverse logistics.

9. I suggest to the authors: a review of native English, re-reading of the article to better explain the correlation between e-waste disposal policies in Vietnam and the focus groups analysed.

Author Response

Dear Reviewer 1, 

Thank you very much for your comments and suggestions. Please find attached our response to your questions.

Best regards,

Reviewer 2 Report

The article is generally well written and deals with important issue. The authors used a variety of statistical tools to reveal the factors affecting residents’ recycling behavior. I have just a few remarks.

1.      The clarity of the paper would increase if the authors provided a short presentation of the structure of their work in the introduction section. The successive steps and aims should be briefly listed.

2.      Although, the used statistical methods are well recognized, a sentence introducing the general idea of each of them would increase the clarity of the paper.

3.      Page 4, line 180 – The sentence ‘Vietnam is …’ seems to be awkwardly formulated.

4.      The same – page 5, lines 199 – 200, phrase ‘ … encompasses a both challenges …’

5.      The same – page 7, lines 311 – 312, sentence ‘In addition, Hansmann ….’

6.      Page 16, lines 551-554 – is any explanation for such difference between China and Vietnam?

7.      Page 18, line 648 – typo in the word ‘Aother’

Author Response

Dear Reviewer 2, 

Thank you very much for your comments and suggestions. Please find attached our response to your questions.

Best regards,

Round  2

Reviewer 1 Report

In the face of all the corrections, I suggested the publication of the article.

As for the difficulty of writing in English by Orientals, do not worry, almost everyone who has not been born in English-speaking countries has the same challenges, even the most experienced in writing in English.

I wish harmony to the authors.